# Improved Assembly of Metagenome-Assembled Genomes and Viruses in Tibetan Saline Lake Sediment by HiFi Metagenomic Sequencing

Ye Tao,[a,b] Fan Xun,[a,b] Cheng Zhao,[a,b] Zhendu Mao,[a,b] Biao Li,[a] Peng Xing,[a] Qinglong L. Wu[a,b,c]

aState Key Laboratory of Lake Science and Environment, Nanjing Institute of Geography and Limnology, Chinese Academy of Sciences, Nanjing, China
bUniversity of Chinese Academy of Sciences, Beijing, China
cCenter for Evolution and Conservation Biology, Southern Marine Sciences and Engineering Guangdong Laboratory (Guangzhou), Guangzhou, China

**ABSTRACT** With the development and reduced costs of high-throughput sequencing technology, environmental dark matter, such as novel metagenome-assembled genomes (MAGs) and viruses, is now being discovered easily. However, due to read length limitations, MAGs and viromes often suffer from genome discontinuity and deficiencies in key functional elements. Here, by applying long-read sequencing technology to sediment samples from a Tibetan saline lake, we comprehensively analyzed the performance of high-fidelity (HiFi) reads and the possibility of integration with short-read next-generation sequencing (NGS) data. In total, 207 full-length nonredundant 16S rRNA gene sequences and 19 full-length nonredundant 18S rRNA genes were directly obtained from HiFi reads, which greatly surpassed the retrieval performance of NGS technology. We carried out a cross-sectional comparison among multiple assembly strategies, referred to as 'NGS', 'Hybrid (NGS+HiFi)', and 'HiFi'. Two MAGs and 29 viruses with circular genomes were reconstructed using HiFi reads alone, indicating the great power of the 'HiFi' approach to assemble high-quality microbial genomes. Among the 3 strategies, the 'Hybrid' approach produced the highest number of medium/high-quality MAGs and viral genomes, while the ratio of MAGs containing 16S rRNA genes was significantly improved in the 'HiFi' assembly results. Overall, our study provides a practical metagenomic resolution for analyzing complex environmental samples by taking advantage of both the short-read and HiFi long-read sequencing methods to extract the maximum amount of information, including data on prokaryotes, eukaryotes, and viruses, via the 'Hybrid' approach.

**IMPORTANCE** To expand the understanding of microbial dark matter in the environment, we did the first comparative evaluation of multiple assembly strategies based on high-throughput short-read and HiFi data from lake sediments metagenomic sequencing. The results demonstrated great improvement of the 'Hybrid' assembly method (short-read next-generation sequencing data plus HiFi data) in the recovery of medium/high-quality MAGs and viral genomes. Further analysis showed that HiFi data is important to retrieve the complete circular prokaryotic and viral genomes. Meanwhile, hundreds of full-length 16S/18S rRNA genes were assembled directly from HiFi data, which facilitated the species composition studies of complex environmental samples, especially for understanding micro-eukaryotes. Therefore, the application of the latest HiFi long-read sequencing could greatly improve the metagenomic assembly integrity and promote environmental microbiome research.

**KEYWORDS** HiFi long-read technology, MAGs, virome, hybrid assembly, 16S/18S rRNA gene, saline lake sediment

Address correspondence to Peng Xing, pxing@niglas.ac.cn.

The authors declare no conflict of interest.

With the rapid development of high-throughput short-read DNA sequencing technologies, generally referred to as next-generation sequencing (NGS), many novel species have been discovered in various habitat contexts, such as human-associated

(1, 2), animal-associated (3, 4), and environmental samples (5, 6). Moreover, the genomes of unculturable species have been assembled at a large scale based on advances in binning bioinformatics tools (7–9). Many successfully recovered metagenome-assembled genomes (MAGs) can not only provide genes/enzymes with novel functions and/or physiochemical properties (3, 10) but also shed light on the special metabolic pathways associated with extreme habitats on Earth (11, 12). The performance of MAG binning depends on the complexity of the habitats, the presence or absence of closely related strains within the metagenomic data set (13), and different binning algorithms (14, 15). The recovery of nearly complete or complete MAGs is the key step in reliable downstream analyses with objectives such as the identification of genes related to specific metabolism (16), novel species identification (17), the evolutionary mechanisms underlying horizontal gene transfer (HGT) (18), and prokaryotic intrapopulation diversity quantification (19). However, many intermediate- or even high-quality MAGs with estimated high completeness and low contamination are often discontinuous and chimeric because of repetitive sequences and mobile gene elements (7). In addition, a great challenge related to MAG construction is the lack of 16S rRNA gene sequences because of the great difficulty in assembling sequences with high similarity (20). An investigative report showed that only 7% of MAGs contained 16S rRNA genes, and it was difficult to link MAGs to the large body of 16S rRNA-based microbiome studies conducted around the world (21). These shortcomings of MAGs generated by short-read sequencing technology in genome assembly limit their application in further metagenome analyses (19).

Compared to next-generation sequencing (NGS) technology based on short-reads (usually less than 300 bp), typically covering only partial gene fragments, third-generation sequencing technologies (TGS) can produce much longer reads, often spanning multiple genes and intergenic regions. Pacific Biosciences (PacBio) and Oxford Nanopore Technologies (ONT) have been the most popular long-read sequencing platforms recently. The development of TGS and corresponding advanced bioinformatics tools, along with affordable sequencing costs, has revolutionized *de novo* genome assembly, which may now retrieve the complete circular genomes of prokaryotes (22). These technologies have also increased the contiguity of assemblies of complex eukaryotic genomes by orders of magnitude, with read lengths exceeding 10 kbp (23). Introducing TGS into metagenomic studies would provide an advantage in contig assembly based on much longer reads (22, 24–26). For example, 20 circularized complete MAGs (cMAGs) from 13 human stool samples were assembled using ONT long-reads, which provided opportunities to investigate the potential functions of repeat elements, although these cMAGs showed lower nucleotide accuracy (24). Moreover, hybrid approaches combining NGS and TGS technologies to perform metagenomic assembly present high potential, including the generation of longer contigs and more complete MAGs for various habitats, such as the human gut, sludge, and soil (25–27). Additionally, genetic variations among microbial genomes in the gut have been identified by hybrid approaches and compared with data from healthy humans (22).

More recently, PacBio high-fidelity (HiFi) read sequencing technology has become popular for assembling complex animal and plant genomes (28, 29). In contrast to the previously available TGS long-reads with a high base error rate, the base-level resolution of HiFi reads is greater than 99%, and assemblies based on HiFi reads include considerably fewer errors at the level of single nucleotides and small insertions and/or deletions than those obtained with ONT technology (30). A total of 428 MAGs with more than 90% completeness, including 44 cMAGs, were recovered from sheep fecal metagenomes using HiFi long reads in a previous study (31). The hybrid strategy of metagenome assembly has been systematically evaluated by assessing the quality of metagenomic sequences (27, 32). However, these evaluations have been focused on the quality of the assembled contigs and gene completeness. The quality of MAGs and the resolution of taxonomic classification based on read levels have been poorly investigated. Moreover, 'hifiasm-meta', a recently released specialized HiFi long read assembler, was shown to be able to reconstruct hundreds of complete circular bacterial genomes based on seven

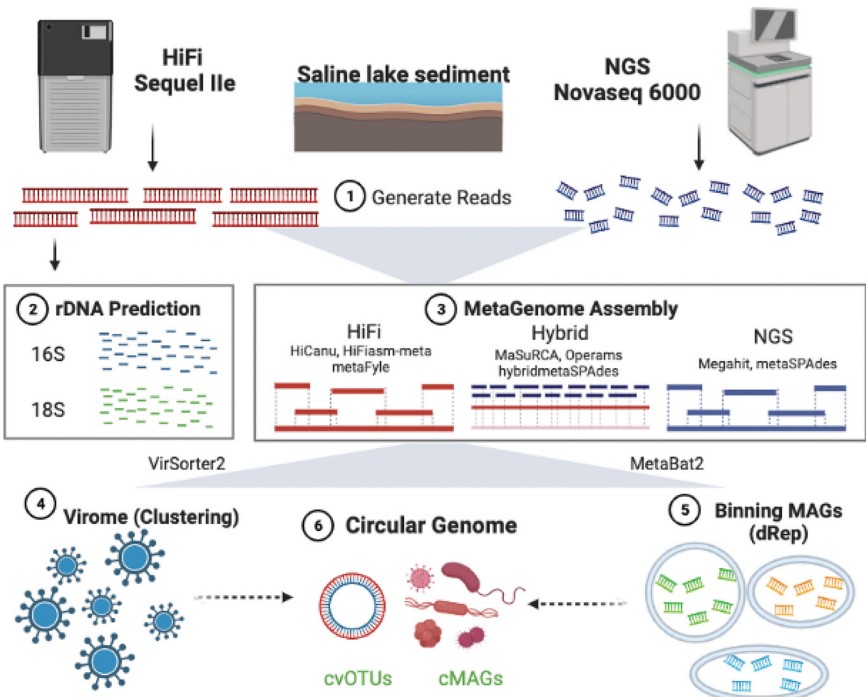

**FIG 1** Flow chart of data analysis. (1) Sediment sampling and sequencing read generation (Long-read: HiFi sequencing data. Short-read: Illumina NGS sequencing data). (2) 16S/18S rRNA gene extraction from HiFi reads. (3) Sequencing data were assembled by 3 strategies, including 'HiFi' (tools: HiCanu, HiFiasm-meta, and metaFyle), 'Hybrid' (tools: MaSuRCA, hybridmetaSPAdes, and Operams), and 'NGS' (tools: Megahit and metaSPAdes). (4) Viral genome identification by Virsorter2. (5) MAG construction and dereplication (ANI > 95%). (6) Circular complete virus and MAG identification.

empirical data sets (33). The performance of HiFi read assemblers and 'NGS + HiFi' hybrid approaches need to be reassessed in complex environmental samples.

Here, we applied NGS (Illumina NovaSeq 6000) and TGS (PacBio Sequel IIe, HiFi sequencing mode) sequencing platforms to perform sequencing in sediment samples from Lake Cuochuolong (CCL, 29°07′45″ N, 85°24′ 11″ E), an alkaline saline lake (Table S1 and Fig. S1) located on the Tibetan Plateau (34). As shown in Fig. 1, metagenomes were assembled with 8 different assemblers, including 3 independent HiFi assemblers (HiCanu, HiFiasm-meta, and metaFyle), 3 hybrid assemblers (MaSuRCA, hybridmetaSPAdes, and Operams), and 2 individual NGS assemblers (Megahit and metaSPAdes with NGS mode). The main objectives of this study were to: (i) evaluate the taxonomic classification abilities of the NGS and HiFi platforms; (ii) comprehensively evaluate the quality of MAGs generated via the NGS, HiFi, and hybrid approaches; and (iii) compare the advantages of virome studies and biosynthetic gene cluster (BGC) detection among different assembly strategies for metagenomes from lake sediment.

## RESULTS

**Sequencing statistics and taxonomic classification.** Two types of sequencing technologies (NGS and HiFi) were applied to obtain metagenomic information for the sediment samples from Lake Cuochuolong. We obtained 39.98 Gbps of paired-end short-read data and 9.17 Gbps of HiFi data, with an average read length of 4,510 bp and an average base accuracy of more than 99% (also called HiFi Q20 reads). There was no PCR step in HiFi sequencing library construction, and the duplication rate of HiFi Q20 reads was 2.1%. A total of 28.9% of the NGS reads could be assigned to known taxonomic ranks, while the read assignment rate of HiFi data was 90.8% (1.85 million reads), which was approximately three times higher than that of NGS (Fig. 2A). The species level assignment rate of HiFi reads was 67.19% (1,366,306/2,033,496), while the corresponding rate of short-reads was only 5.59%, indicating the great power of

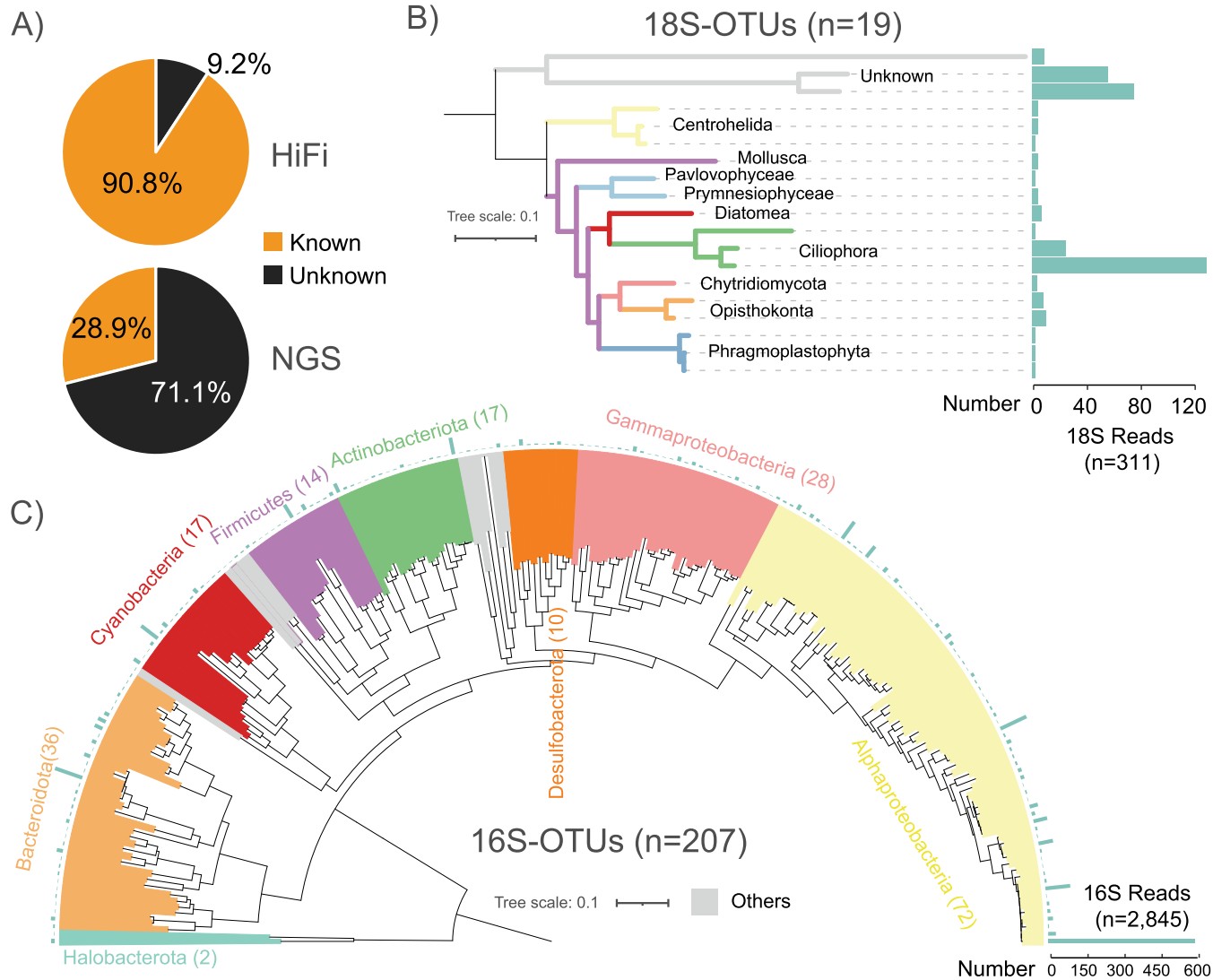

**FIG 2** Taxonomic resolution based on HiFi reads. (A) Percentage of HiFi and NGS reads with (known)/without (unknown) taxonomic information. (B and C) Phylogenetic trees based on 19 full-length 18S-OTUs (B) and 207 full-length 16S-OTUs (C).

HiFi reads in taxonomic classification. A total of 17 phyla with a minimum relative abundance greater than 0.1% were identified, and Alphaproteobacteria was the most abundant group in this sediment sample (Fig. S2). Although there were large differences in the ratios of reads with taxonomic information, a comparison of the community composition at the phylum level revealed high overall similarity (correlation coefficient = 0.988) between the short-read and long-read data sets (Fig. S3).

Since the average length of the HiFi reads was longer than that of 16S/18S rRNA genes, we extracted complete 16S/18S sequences directly from the read data set. For eukaryotes, 311 complete 18S rRNA sequences were successfully extracted, and most of them (97.43%, 303/311) were novel at the species level. Nineteen 18S operational taxonomic units (18S-OTUs, > 98.65% identity, and > 95% coverage cutoff) (35) were generated. Based on these 18S-OTUs, we assessed the eukaryote distribution in saline lake sediment, which included the phyla Ciliophora (44.05%), Opisthokonta (5.14%), Centrohelida (3.21%), Diatomea (1.93%), Phragmoplastophyta (1.93%), Mollusca (1.29%), Prymnesiophyceae (1.29%), Chytridiomycota (0.96%), and Pavlovophyceae (0.64%) (Fig. 2B). One 18S-OTU affiliated with phylum Ciliophora, subclass Hypotrichia, was the most abundant eukaryotic species (36.33%, 113/311) in the CCL sediment sample. Three 18S-OTUs with unknown taxonomic information were also abundant (39.55%, 123/311). Additionally, 2,845 complete 16S rRNA

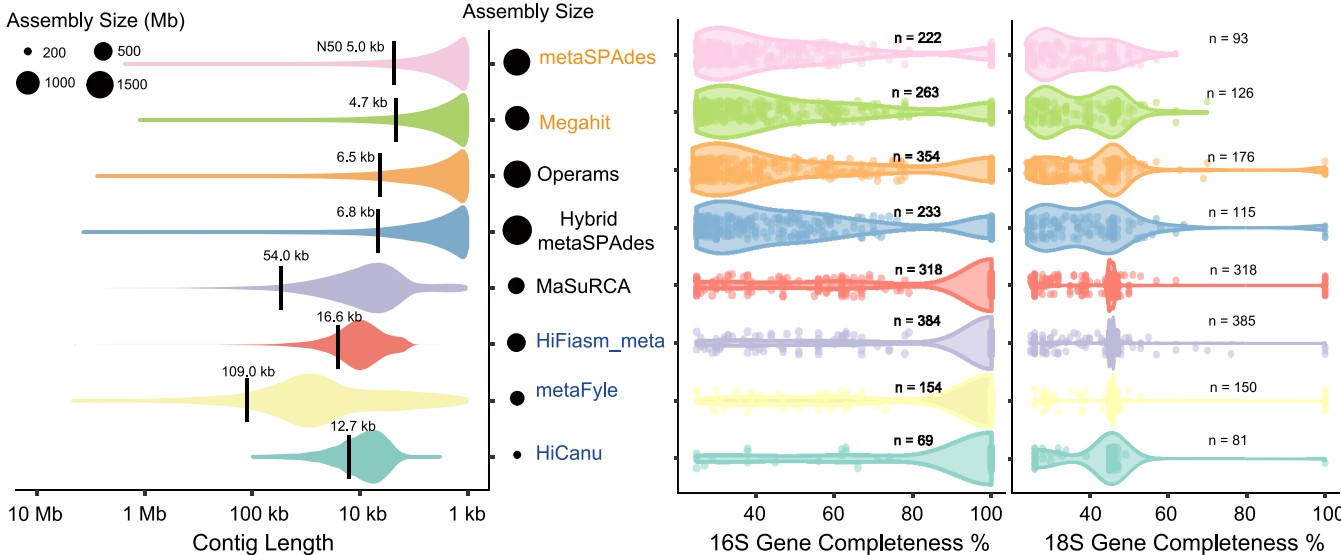

**FIG 3** Summary of the statuses of the eight assemblies from Lake CCL sediment. The completeness of the 16S/18S rRNA genes was calculated with Barrnap software.

sequences were successfully extracted from the HiFi reads, among which one quarter (726/2,845) of the 16S rRNA genes could be assigned to known species, and 74.48% (2,119/2,845) and 5.59% (159/2,845) were novel at the species and family levels, respectively. A total of 207 16S-OTUs were generated, spanning 1 archaeal phylum (two 16S-OTUs) and 13 bacterial phyla (Fig. 2C). The most abundant 16S-OTU was identified as *Loktanella* sp. (20.74%, 590/2,845), classified in Alphaproteobacteria.

**Sequence assembly programs.** In this study, we applied 3 different strategies to assemble contigs, categorized as the 'NGS', 'Hybrid', and 'HiFi' modes, with 8 assemblers (Fig. 3). The programs HiCanu, HiFiasm-meta, and metaFyle were used to assemble the PacBio HiFi reads and generate contigs with sizes ranging from 56.21 Mb to 505.17 Mb (contigs > 500 bp) and N50 values ranging from 12.68 kb to 109.02 kb (Fig. 3 and Table 1). The average contig size generated with HiCanu was extremely small, corresponding to approximately 25% of the sizes obtained from the other 2 'HiFi' assemblers. However, the mapping rate of NGS reads to HiCanu assemblies was approximately 50% of the rates for the HiFiasm-meta and metaFyle assemblies, indicating that HiCanu might only generate microbial genomes with a relatively high abundance in the sediment sample. The performance of multiple metagenomic assemblers supporting long-read data exhibited huge differences and the same phenomenon was also found in human fecal samples indicating a lack of widely accepted analysis pipelines currently (36). Additionally, 248 circular contigs (> 5 kb), ranging from 5 kb to 3.03 Mb in size, were produced with the HiFiasm-meta and metaFyle. The programs MaSuRCA, hybridmetaSPAdes, and Operams were used to assemble short-read and long-read data simultaneously and generated

**TABLE 1** Statistic summary of contigs produced by the eight assemblers

| | HiFi | | | Hybrid | | | NGS | |
|---|---|---|---|---|---|---|---|---|
| Assembler | HiCanu | metaFlye | HiFiasm_meta | MaSuRCA | HybridmetaSPAdes | Operams | Megahit | metaSPAdes |
| ContigNumber | 4813 | 4611 | 33749 | 17008 | 3283225 | 1762307 | 712192 | 3446523 |
| Largest contig(Mb) | 0.10 | 4.66 | 4.51 | 8.68 | 3.72 | 2.77 | 1.11 | 1.53 |
| ContigNumber(>100kb) | 0 | 420 | 215 | 515 | 496 | 254 | 215 | 266 |
| Total size(Mb) | 56.21 | 229.77 | 505.17 | 352.47 | 1828.59 | 1486.96 | 1099.89 | 1795.67 |
| N50(kb) | 12.68 | 109.02 | 16.57 | 53.97 | 6.81 | 6.54 | 4.69 | 4.97 |
| N90(kb) | 6.51 | 24.03 | 7.52 | 7.83 | 1.37 | 1.35 | 1.27 | 1.26 |
| GC% | 60.79 | 55.94 | 57.06 | 54.00 | 51.93 | 51.84 | 51.45 | 51.29 |
| NGS-reads-mapping rate%[a] | 25.52 | 59.04 | 55.45 | 64.87 | 83.13 | 82.48 | 81.77 | 83.03 |

[a]Detailed calculation of 'NGS-reads-mapping rate%' was introduced in the method section.

contigs with sizes ranging from 352.47 Mb to 1,828.59 Mb (contigs > 500 bp) and N50 values ranging from 6.54 kb to 53.97 kb (Table 1). The programs Megahit and metaSPAdes were used to assemble NGS data and generated shorter contigs with sizes ranging from 1,099.89 Mb to 1,795.67 Mb (contigs > 500 bp) and N50 values ranging from 4.69 kb to 4.97 kb. The MaSuRCA and hybridmetaSPAdes pipelines produced a large number of contigs > 100 kb (Table 1).

Thus, the 'HiFi' methods produced much longer contigs than the other 2 strategies; however, the assembly sizes of 'HiFi' contigs were much smaller, accounting for only 25.52 to 59.04% of the NGS read mapping rates. among the 'Hybrid'-assembled results, MaSuRCA was prominent, with a maximum contig length of 8.68 Mb, and had a higher N50 than the HiCanu and HiFiasm-meta 'HiFi' assemblers. In general, when longer contig lengths were generated, smaller contig sizes were obtained accordingly (Fig. 3). Detailed statistics of the various assembly evaluations are listed in Table 1.

We also investigated the occurrence of 16S/18S rRNA sequences in the assembled contigs. HiFiasm-meta generated the highest number of 16S rRNA genes and completed 16S rRNA numbers, followed by MaSuRCA and Operams. Overall, HiFiasm-meta and MaSuRCA showed outstanding performance for 16S/18S rRNA gene detection (Table S2 and Fig. 3). After dereplication, 130 complete 16S-OTUs and 9 complete 18S-OTUs were identified from eight assembled contigs, including 14 prokaryotic phyla and 7 eukaryotic phyla, largely matching the taxonomic results obtained with HiFi reads (Table S2). According to the 207 complete 16S-OTUs and 19 complete 18S-OTUs identified directly from HiFi reads, MaSuRCA recovered 62.80% of the 16S-OTUs (130/207), and both HiFiasm-meta and metaFlye recovered 36.84% of the 18S-OTUs (7/19) (Table S2). Based on the comprehensive evaluation of the N50 lengths, read usage ratios, and contig sizes of the assemblies and the identification of complete 16S/18S rRNA sequences, we suggest the application of multiple assembly tools simultaneously to improve the assembled metagenome quality.

**MAG retrieval and comparison of MAG characteristics.** Initially, 1,215 raw MAGs were individually obtained from 8 assemblies, including 184 high-quality (>90% completeness and <5% contamination), 242 medium-quality (≥ 50% completeness and <10% contamination) and 789 low-quality MAGs. Among the 8 programs, 'Hybrid' metaSPAdes produced the highest number of MAGs (Table 2). Considering the potential errors caused by incompleteness and contaminant sequences in MAGs, low-quality MAGs were discarded from further analysis. Then, 426 high/medium-quality MAGs were used to determine the performance of different assembly strategies. Thus: (i) The MAG size ranged from 675 kb to 8.76 Mb, with an average of 2.87 Mb. There was no statistically significant difference in genome size (Fig. 4A) or heterozygosity (Fig. 4B) among the 3 groups. (ii) The GC content was significantly different between 'NGS'- and 'HiFi'-generated MAGs (Fig. 4C), but no significant difference was found between the 'Hybrid' assemblies and the others. (iii) The evaluation of genome continuity showed significant differences; for example, the average contig number of 'NGS' MAGs was 239, which was more than three times larger than that of 'HiFi' MAGs (Fig. 4D). (iv) The N50 length of 'HiFi' MAGs was approximately 458 kb, which was > 10 times that of 'NGS' MAGs (Fig. 4E). (v) Although the average contig number of the 'Hybrid' assemblies was 180, which was > 3 times that of 'HiFi' MAGs, the N50 length was close to half of that of the 'HiFi' MAGs. (vi) In addition, the relative abundance of 'HiFi' MAGs was significantly higher than those of other genomes, and there was no significant difference in relative abundance between 'NGS' and 'Hybrid' MAGs (Fig. 4F).

Thereafter, 426 high/medium-quality MAGs were clustered into 102 representative MAGs (>95% ANI) spanning 11 phyla according to the GTDB-TK taxonomic classification system (Table S3 and Fig. 5A). The most commonly identified taxa were members of Proteobacteria (43.14%), including Alphaproteobacteria (31.37%), and Gammaproteobacteria (11.76%), followed by members of Bacteroidota (22.55%), Firmicutes (13.73%) and Actinobacteriota (5.88%). Twenty-one MAGs were found to be novel at the genus level, and the most abundant MAG was affiliated with *Loktanella* spp., in accord with the 16S-OTU results (Fig. 2C and 5A). Among the 102 representative MAGs, the 'Hybrid' strategy exhibited the best

**TABLE 2** Statistic summary of MAGs generated by the eight assemblers

| Assembler | HiFi | | | Hybrid | | | NGS | |
|---|---|---|---|---|---|---|---|---|
| | HiCanu | metaFlye | HiFiasm_meta | MaSuRCA | HybridmetaSPAdes | Operams | Megahit | metaSPAdes |
| High-quality | 0 | 16 | 8 | 15 | 41 | 30 | 37 | 37 |
| Medium-quality | 1 | 19 | 15 | 36 | 43 | 49 | 40 | 39 |
| Low-quality[a] | 14 | 61 | 132 | 91 | 144 | 131 | 106 | 110 |
| GenomeSize (Mb) | 2.61 | 0.86 to 8.71 | 1.68 to 8.64 | 0.74 to 5.08 | 0.68 to 6.10 | 0.61 to 8.49 | 0.68 to 8.76 | 0.64 to 8.81 |
| No. of Contigs in one genome | 166 | 1 to 116 | 2 to 191 | 1 to 214 | 6 to 968 | 9 to 693 | 26-962 | 12 to 849 |
| N50(kb) | 17.38 | 34.06 to 4657.02 | 13.09 to 4511.23 | 15.49 to 3805.86 | 5.02 to 1110.41 | 4.81 to 2765.81 | 3.97-420.02 | 4.64 to 357.97 |
| No. of MAGs with Complete 16S rRNA genes | 1 | 25 | 18 | 31 | 20 | 21 | 14 | 10 |
| No. of representative MAGs | 1 | 35 | 23 | 51 | 84 | 79 | 77 | 76 |

[a]Low-quality MAGs were not used in further statistics.

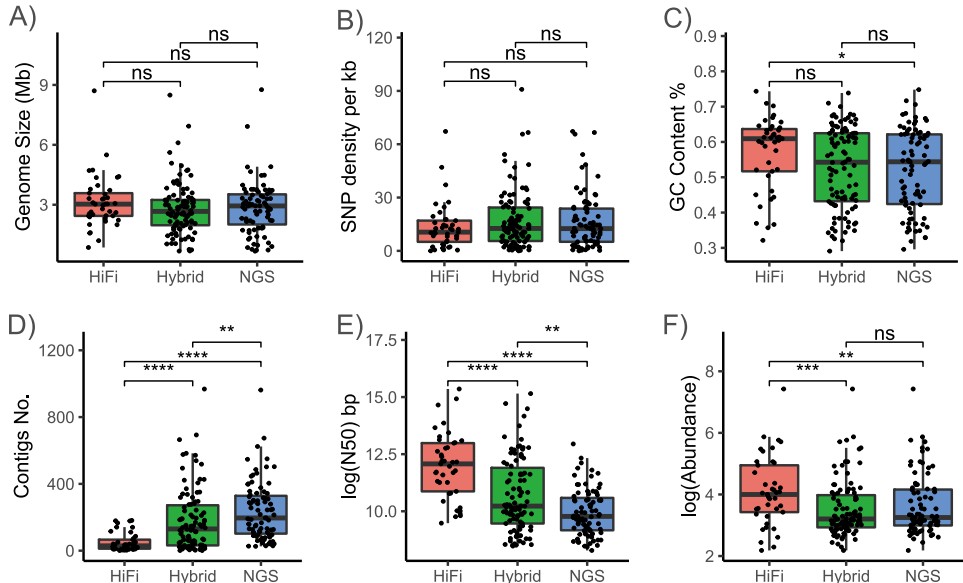

**FIG 4** Comparison of MAG characteristics based on 3 types of strategies, 'HiFi', 'Hybrid' and 'NGS', for genome size (A), the genome-wide ratio of the number of heterozygous calls (B), GC content (C), contig number (D), N50 length (E), and relative abundance (F). *P*-values are calculated from t-tests. *, $P < 0.05$; **, $P < 0.01$; ***, $P < 0.001$; ****, $P < 0.0001$; ns, not significant.

performance, recovering the majority of the representative MAGs (96.1%) (Fig. 5B). Only one-third (34/102) of the MAGs could be recovered by all assemblers. Moreover, most high/medium-quality 'NGS' MAGs (84.3%, 129/153) lacked complete 16S rRNA sequences. In contrast, 90% (54/60) of the high/medium-quality 'HiFi' MAGs contained full-length 16S rRNA elements, and 38.3% (23/60) even included more than one copy. Among 214 high/medium-quality 'Hybrid' MAGs, 29 (13.6%) MAGs contained at least 2 copies of 16S rRNA, and 56 (26.2%) had a single copy (Fig. 5C).

**Complete MAG (cMAG) assembly ability.** Twenty-four circular contigs (excluding plasmids and viruses) larger than 100 kb were first filtered with the 'HiFi' programs HiFiasm-meta (16/24) and metaFyle (8/24), whereas no circular contigs were identified by HiCanu. After further MAG construction and genome reduplication, 2 representative cMAGs including all types of rRNA elements were successfully recovered from the sediment samples (Fig. 1) (see methods for more details).

The first cMAG 'ccl-hifiasmmeta.153' aligned with the genus *Psychroflexus* in the family Flavobacteriaceae. *Psychroflexus* sp. is a Gram-negative, nonmotile, obligatory aerobic bacterium that is frequently identified in saline environments for which there is currently no complete genome available in the NCBI GenBank database (37, 38). Compared with 24 known draft genomes of *Psychroflexus* spp. (Fig. S4), 'ccl-hifiasm-meta.153' showed a relatively divergent genome (ANI < 85%) except strain CCL10W[T] (GCF_020164535.1, ANI > 99.99%, 15 scaffolds), which was isolated from the same lake (Table S4). These 2 genomes were nearly identical according to genome-wide collinearity analysis (Fig. 6A). Remarkably, unlike most bacterial genomes, 'ccl-hifiasmmeta.153' had 2 circular chromosomes (Fig. S5A) with genome sizes of 2.56 Mb and 463.56 Kb, respectively. Moreover, chromosome I contained the replication initiator protein *DnaA* (Fig. S5A).

The second cMAG, 'ccl-flye.84', was identified as a novel *Methylovirgula* species based on the GTDB-Tk toolkit with an ANI > 95% cutoff (Table S4). Comparison with known species from the NCBI RefSeq database revealed that the most closely related species (ANI = 82.09%) was *Methylovirgula ligni* (Fig. 6B), which is a methanotroph with high abundance in tropical peat domes (39). *M. ligni* can utilize methane (CH$_4$) and might be a factor contributing to low CH$_4$ gas emissions (40). Therefore, we investigated the genes involved in methane metabolism (KEGG: ko00680). In contrast to the most closely related genomes,

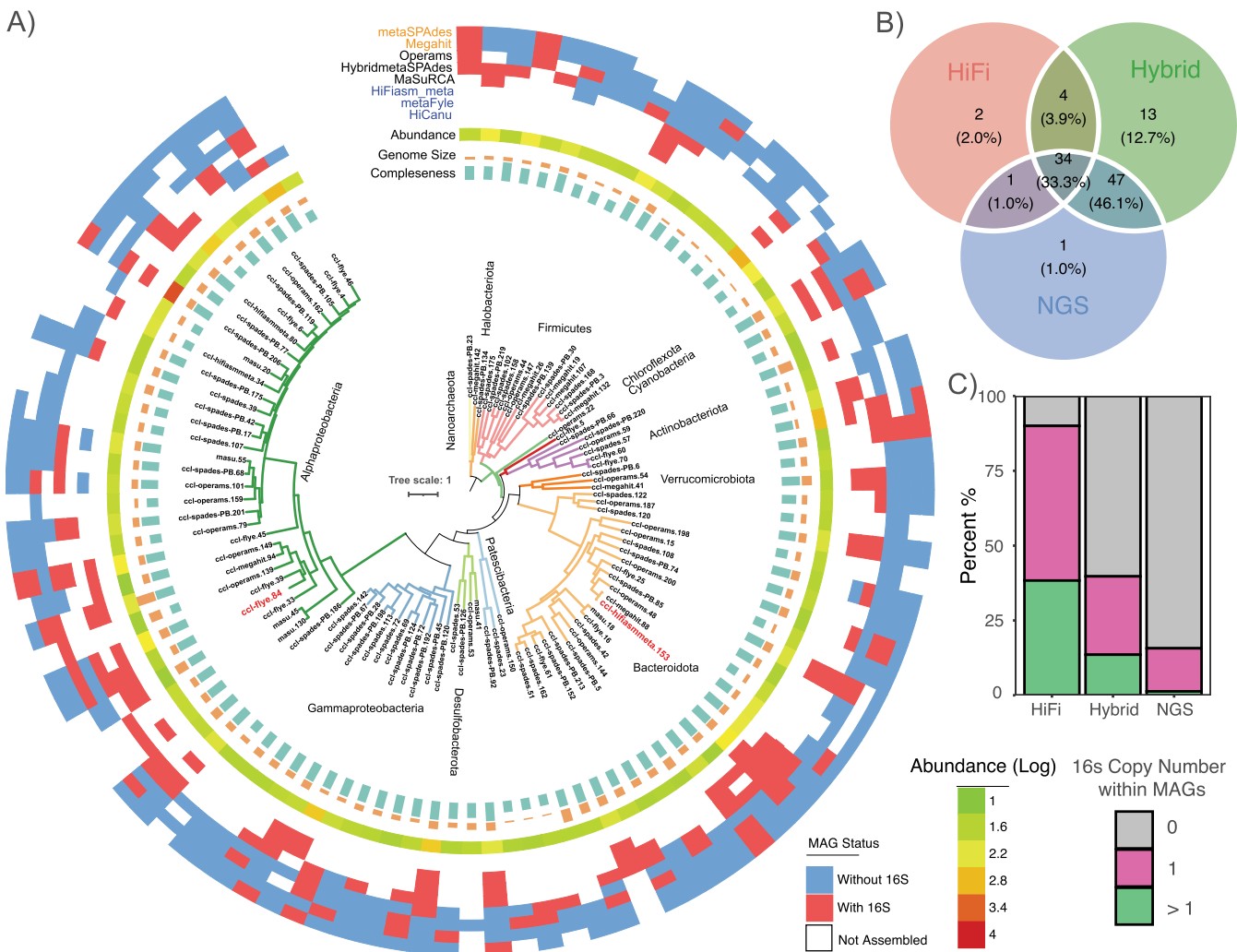

**FIG 5** Phylogeny of representative MAGs identified from Lake CCL sediment. (A) Phylogenetic trees based on PhyloPhlAn 3.0 marker genes from 102 representative MAGs. MAG completeness, genome size, relative abundance, and assembly sources are presented from the inner to the outer circle. Colored branches correspond to phyla inferred with GTDB-Tk. The number of representative MAGs (B) and the 16S rRNA copy number distribution (C) are shown for the 3 assembly strategies.

'ccl-flye.84' lacked *mxaACGKL* proteins, which are encoded by large gene clusters involved in methanol oxidation (41), indicating that 'ccl-flye.84' might have lost the capacity for methanol oxidation (Fig. 6B). In addition, 'ccl-flye.84' contained the *dmd-tmd* [EC:1.5.8.1], *fdhA* [EC:1.2.1.46], and *AGXT* [EC:2.6.1.44] genes, which were absent in the *M. ligi* genomes.

**Identification of viral genomes.** A total of 1,486 putative viral genomes ($> 10$ kb) were identified from the 8 assemblies (Table 3). All viral genomes were classified as double-stranded DNA phages, and most of the (97.11%, 1,443/1,486) were linear. The viral genomes from the 'Hybrid' assemblies ranged from 10 kb to 331.4 kb, with an average size of 31.9 kb; these genomes were significantly longer than those obtained via the 'NGS' (average size: 27.7 kb) approach and nearly identical to those obtained via the 'HiFi' approach (average size: 31.4 kb) (Fig. 7A). GC contents (Fig. 7B) and relative abundance (Fig. 7C) significantly differed among the 3 groups. The viral genomes obtained from the 'HiFi' assemblies showed higher GC contents than those from 'Hybrid' and 'NGS' assemblies. Similar to the findings for MAGs, the 'NGS' methods can identify more low-abundance viral genomes than the other methods. In addition, the genome completeness patterns of the obtained viral genomes were similar between the 'HiFi' and 'Hybrid' results, while the ratio of low-quality viral genomes was highest in the 'NGS' results (Fig. 7D). Moreover, 506 viral operational taxonomic units (vOTUs)

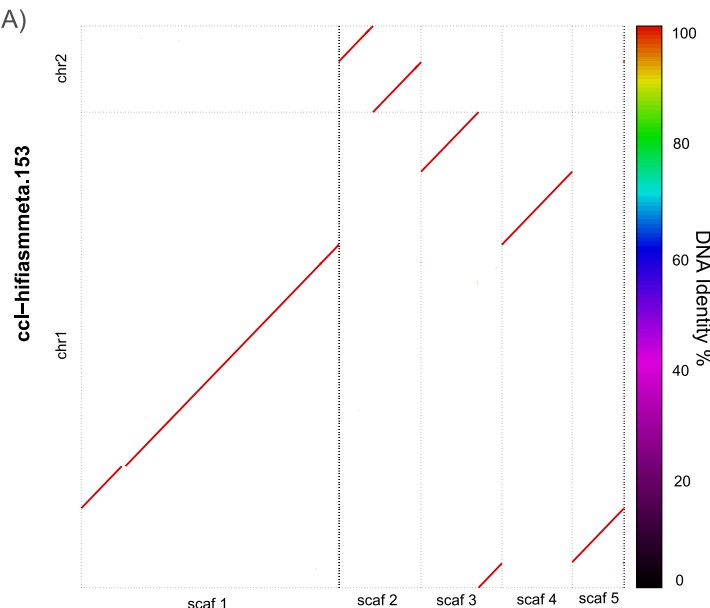

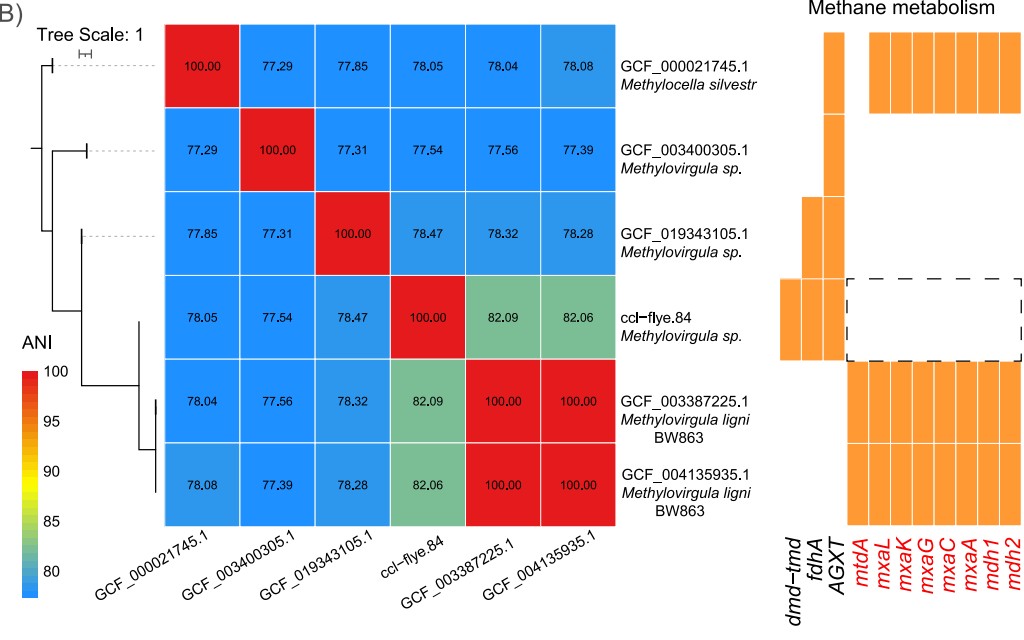

**FIG 6** Comparative genome analysis of two complete MAGs (cMAGs). (A) Genome alignment plot between cMAG ccl−hifiasmmeta.153 (*y* axis) and the most closely related strain CCL10W$^T$ (*x* axis). (B) ANI values among cMAG ccl−flye.84 and the most closely related published genomes from the NCBI RefSeq database. Absent/present genes in the methane metabolism pathway are shown in the right panel.

were clustered according to MIUViG standards (42). Consistent with the status of MAG recovery, the 'Hybrid' strategy exhibited the best performance, recovering the majority of the representative vOTUs (84.6%) among the 506 representative vOTUs (Fig. 7E and Table S5). Among the 8 assemblies, the 'hybrid' tool MaSuRCA generated 176 (34.8%) vOTUs and was the most important contributor to vOTU identification, followed by hybridmetaSPAdes (116, 22.9%) and HiFiasm-meta (72, 14.2%) (Fig. 7F).

To assess the advantages of long-read sequencing technology in virome research, 29 complete viral operational taxonomic units (cvOTUs), including 43 circular viral genomes, were subjected to further analysis (Fig. 7F). The length of the cvOTUs ranged from 24.4 kb to 331.5 kb, and 3 dominant viral families, Siphoviridae, Podoviridae, and

**TABLE 3** Statistic summary of viral genomes detected by the eight assemblers

| Assembler | HiFi | | | Hybrid | | | NGS | |
|---|---|---|---|---|---|---|---|---|
| | HiCanu | metaFlye | HiFiasm_meta | MaSuRCA | HybridmetaSPAdes | Operams | Megahit | metaSPAdes |
| High-quality[a] | 0 | 32 | 11 | 44 | 25 | 13 | 10 | 16 |
| Medium-quality | 1 | 35 | 34 | 70 | 47 | 23 | 28 | 24 |
| Low-quality | 7 | 42 | 178 | 167 | 182 | 165 | 154 | 178 |
| No. of viral genomes | 8 | 109 | 223 | 281 | 254 | 201 | 192 | 218 |
| No. of circular genomes | 0 | 29 | 14 | 0 | 0 | 0 | 0 | 0 |
| GenomeSize $\pm$ s.e. (kb) | 15.60 $\pm$ 1.75 | 46.73 $\pm$ 3.75 | 24.53 $\pm$ 1.67 | 35.97 $\pm$ 2.21 | 33.15 $\pm$ 2.09 | 24.71 $\pm$ 1.21 | 27.60 $\pm$ 1.56 | 27.80 $\pm$ 1.97 |

[a]The quality of viral genomes was calculated by CheckV.

Myoviridae, constituted the core circular viromes (Table S6). Remarkably, a total of the 29 cvOTUs compared against known viral databases, such as NCBI RefSeq viral genomes, marine viral database GOV 2.0, and IMG/VR, were found to be novel viruses (Fig. S6). Approximately two-thirds of the cvOTUs did not contain any auxiliary metabolic genes (AMGs), and 18 AMGs identified by VIBRANT were found across 9 cvOTUs (Fig. S7A). The number of AMGs within each cvOTU ranged from 1 to 4, and the longest cvOTU, 'congtig_4947', included 4 AMGs involved in nicotinate/nicotinamide metabolism, methane metabolism and folate biosynthesis (Fig. S7A). We found 2 cvOTUs ('contig_4601' and 'contig_4636') with AMGs related to microbial biogeochemical cycling, and both contained *phnP* (EC 3.1.4.55, 5-phospho-$\alpha$-d-ribose 1,2-cyclic phosphate 2-phosphohydrolase), which was classified as being related to phosphonate/phosphinate metabolism and can participate in the degradation of methylphosphonate. In addition to *phnP*, 'contig_4601' also included a *DNMT3A* gene (EC:2.1.1.37, DNA 5-cytosine methylase), encoding a product involved in methane metabolism (Fig. S7B). 'contig_4636' contained *cysH* (EC:1.8.4.8, phospho-adenosine phosphosulfate reductase), whose product plays a role in the assimilation of sulfate and the catalysis of the reduction of 3′-phospho-adenylylsulfate (PAPS) to sulfite (Fig. S7C) (43).

**Identification of BGCs.** Finally, we investigated the advantage of HiFi sequencing technology in BGC detection. BGCs are notoriously difficult to identify in NGS-based metagenomic studies, mainly due to the need for long gene regions and complex gene structures (44). We identified 2,883 BGCs by using antiSMASH (45), including 1,821 complete ones and 1,062 partial BGCs across the eight assemblies. Most of the BGCs (91.2%) were identified as novel, demonstrating the ability to employ the tested assembly strategies for the exploration of novel natural products (Fig. S8A). The HiFiasm-meta, hybridmetaSPAdes, and Operams assemblies showed the greatest numbers of complete BGCs (Fig. S8B). Regarding partial BGCs, the 'HiFi' and 'Hybrid' assemblies included more BGCs than those generated from 'NGS' methods (Fig. S8C). The 28 complete secondary metabolite BGCs included terpene (30.4%), nonribosomal peptide synthetases (NRPS, 12.3%), hserlactone (12.1%), bacteriocin (10.3%), and polyketide synthase (PKS, T3PKS and T1PKS, 12.3%).

## DISCUSSION

**The advantage of long reads for taxonomic classification.** Our study extracted thousands of complete 16S/18S rRNA gene sequences and determined the number of different microbes in the environmental sample based on whole metagenome sequencing by using HiFi reads directly. To our knowledge, this work represents the first survey of eukaryotic species in saline lake sediment using long-read-based metagenome data (46, 47). We applied the selected method rather than previously applied rRNA-based species community survey technologies for the following reasons: (i) There is no requirement for PCR in HiFi sequencing library construction, while there are at least 2 PCR steps in the NGS sequencing protocol, which introduces community bias caused by high GC contents and complex gene structures (48). (ii) NGS-based rRNA gene recovery requires multiple types of primers to amplify hypervariable regions, and profiling performance varies in different habitats (49). Moreover, rRNA reads from

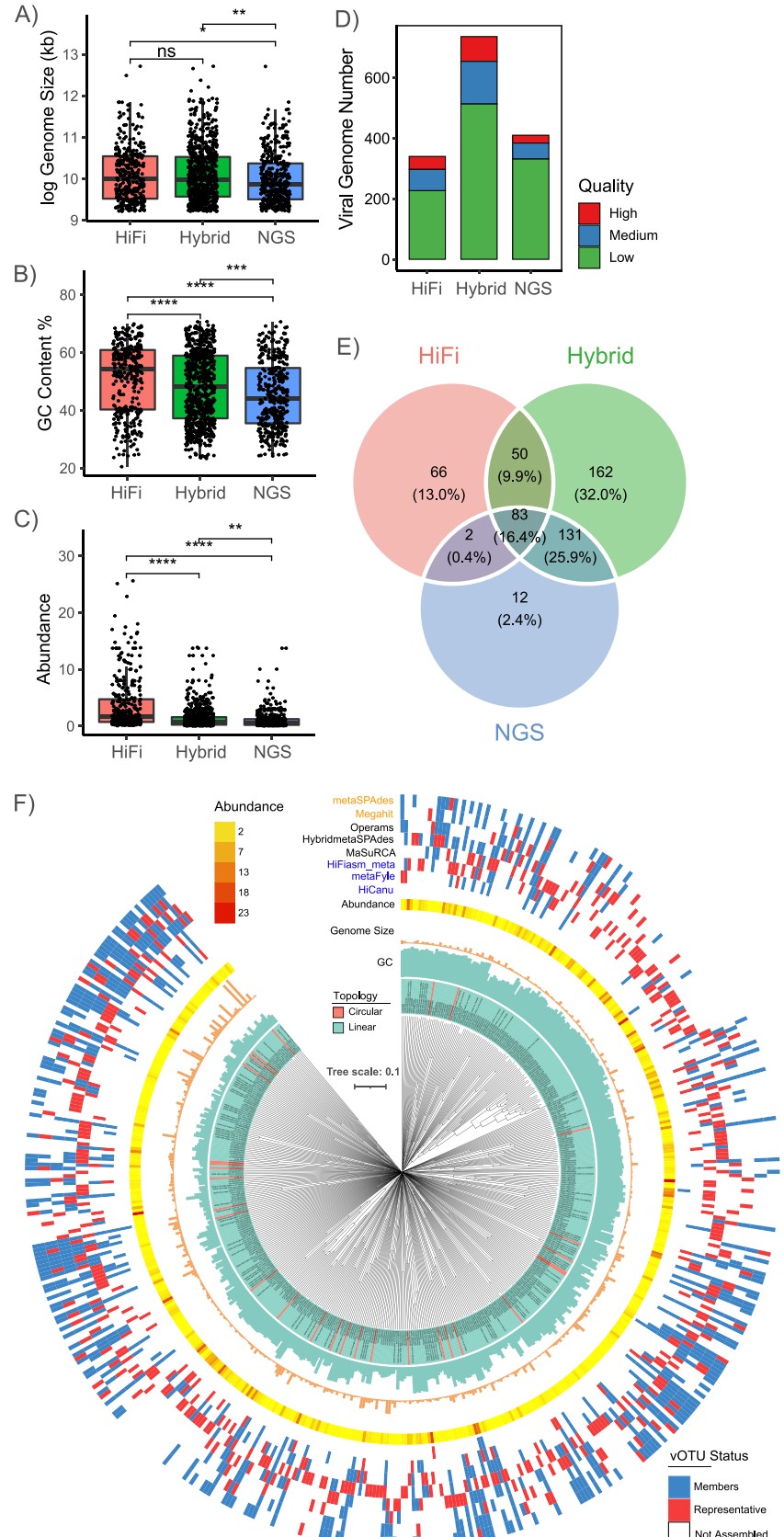

**FIG 7** Genome characteristics and phylogeny of viral genomes from Lake CCL sediment. Genome size (A), GC content (B), and relative abundance (C) were compared among the 'HiFi', 'Hybrid', and 'NGS'

metagenomic studies provide a source of sequences that are not subject to PCR primer bias and therefore cover taxa that might be missed with existing primer sets (50). In contrast, our method extracted rRNA sequences directly from long reads without considering possible suitable primers. (iii) Although most NGS-based metagenomic studies ignore rRNA genes, which are often missing or difficult to recover, it is notable that tools such as REAGO (20) and EMIRGE (51) can reconstruct full-length rRNA genes from short-read sequencing data. Due to the lower gene accuracy and recovery efficiency of such tools, they would not have been popular in previous studies. Within our pipeline, as many full-length 16S/18S rRNA sequences as possible were identified without any assembly process. Of course, the HiFi sequencing depth of samples from different habitats should be evaluated in the future to meet the requirements of various further analyses.

**MAG construction via the hybrid assembly strategy.** Recently, long-read sequences have commonly been combined with short-read data to assemble metagenomes, which are also known as 'Hybrid' assemblies (32, 52). This hybrid sequencing strategy takes advantage of both NGS and TGS and overcomes the limitations of long reads (PacBio CLR or Nanopore platform), which show higher error rates and low throughput. Long-read-only assemblies (specifically based on HiFi reads) of metagenomes are still rare, but the advantages of this method are obvious, including a high recovery rate of complete 16S rRNA genes, longer MAG N50, and higher nucleotide accuracy of MAGs (25). Here, in addition to comparing the quality MAGs from different assembly strategies, we also evaluated the effect factors for MAG construction. First, 'HiFi' contigs recovered more genome regions with a higher GC content, where the performance of NGS technology was relatively poor (Fig. 4C). Next, the intraspecies diversity of bacteria and the heterozygosity rate of eukaryotes are among the most important factors affecting genome assembly quality (53). However, the effect of intraspecies diversity, mainly reflected by SNP density, was not as high as expected in our sample (Fig. 4B). Finally, since the cost of HiFi technology is still much higher ($>$ 10-fold) than that of NGS technology, the amount of our HiFi data was only one quarter the amount of our NGS data. Therefore, the ability to identify more MAGs was highest when applying 'Hybrid' methods (Fig. 5B) and only MAGs with relatively high abundance could be identified in 'HiFi' mode (Fig. 4F). Overall, our results suggested that until HiFi sequencing prices drop significantly, combining 'HiFi' and 'Hybrid' assembly strategies will present obvious advantages in MAG identification and could be the best choice in metagenome studies.

**Improvement of virome and BGC research.** HiFi-based metagenomic technology has resulted in great improvements in the discovery of functional genes, such as viral genes and BGCs. To date, the interpretation of complete viromes has been a significant issue because 75 to 95% of published viral metagenomic reads from the human gut remain unclassified (54, 55), and 58 to 66% of environmental viral genomes are extremely incomplete and defined as low-quality viral genomes by CheckV (5, 42, 56). Moreover, incomplete viral genomes limit host prediction, and MAG and phage-prophage identification (57, 58). Deep metagenomic sequencing would be a plausible solution to this problem, and we analyzed large amounts of short-read data in this study (39.98 Gbps). The estimated coverage of our NGS data (Fig. S9A) and the 'Megahit' ('NGS' method) (Fig. S9B) assembly size were simultaneously close to saturation. However, the corresponding average length of the viral genomes was only 27.7 kb, which was much shorter than the 'HiFi' and 'Hybrid' outputs. Additionally, our pipeline produced complete circular viral genomes, which cannot be achieved based on a

**FIG 7** Legend (Continued)
assembly strategies. (D) The viral genome quality was estimated by CheckV. $P$-values are calculated from t-tests. *, $P < 0.05$; **, $P < 0.01$; ***, $P < 0.001$; ****, $P < 0.0001$; ns, not significant. (E) The number of shared vOTUs among the three assembly strategies. (F) Proteomic trees based on the 1-$S_G$ distance matrix among 506 vOTUs. The vOTU GC content, genome size, relative abundance, and assembly source are shown from the inner to the outer circle. Colored branches correspond to the topology of vOTUs.

short-read data set. Here, the cvOTUs 'contig_4601' and 'contig_4636' contained different P-metabolism genes (*phnP*), indicating that viruses can affect P cycling in lake sediment. Of course, future studies involving HiFi sequencing technology will be needed to investigate the roles of other virus types, including single-stranded DNA and RNA viruses, in lake sediment.

The products of BGCs are sources of antibiotics and cancer therapies, and tens of thousands of novel BGCs have been identified from published MAGs; however, most MAGs have been generated by using short-read NGS technology (59). Due to the high diversity of core genes and relatively high GC content across the PKS/NRPS regions, most novel BGCs could not be extracted efficiently by NGS methods (60). Long-read technologies, such as Nanopore and PacBio sequencing, have been successfully applied to the analysis of soil and animal gut metagenomes to identify BGCs (31, 61). Interestingly, under the 'HiFi' assembly strategy, HiFiasm-meta performed best both in virus and BGC detection, suggesting that this recently designed tool was the most suitable for assembling high-fidelity long reads. A similar phenomenon was not observed under the 'Hybrid' and 'NGS' assembly strategies. Therefore, our results demonstrated that combining HiFiasm-meta, hybridmetaSPAdes, and Operams could generate more complete BGCs than a single assembly tool.

**Conclusions.** The results obtained here demonstrate that there are different benefits of bioinformatical assembly tools in identifying multiple types of genome elements. 'HiFiasm-meta' within the 'HiFi' assembly strategy produced more complete viruses, BGCs, and cMAGs with a genome accuracy almost equivalent to those of single isolated genomes. The quantity and quality of the obtained MAGs were both better under 'Hybrid' approaches. 'NGS' methods generated the maximum gene numbers and contig sizes. Therefore, combining the 'NGS', 'HiFi', and 'Hybrid' assembly strategies could be the best way to extract the maximum information of complex metagenomic samples until the cost of long reads drops significantly. We also revealed that the full-length 16S and 18S rRNA sequences could be extracted directly from HiFi reads to evaluate microbial and eukaryotic diversity, and in the future, this method might be used in ecological community research.

## MATERIALS AND METHODS

**Sample collection and genomic DNA extraction.** The altitude of the alkaline Lake Cuochuolong is 4610 m above sea level and the average water depth was 2.5 m (34). Surface sediment was collected from 5 sites on September 17th, 2021 with a water temperature of 14.4°C and pH of 9.24, respectively (Table S1 and Fig. S1). Five grams of sediment was transferred into the freezing tube and stored at −20°C. Metagenomic DNA extraction was performed using a QIAamp DNA Stool minikit (Qiagen, cat. no. 51604) for each sample and the genomic DNA was mixed for metagenomic sequencing. Because of the extremely high salinity (>100‰) of Lake CCL and the fact that the salt concentration has a great effect on the stability of DNA molecules (62), the length of the DNA fragments, ranging from 200 bp to 8000 bp, was shorter than those from freshwater sediments. We made some minor protocol modifications to extract longer DNA fragments to meet the requirements of long-read sequencing technology. Briefly, in the second step, we followed the major instructions for the 'Isolation of DNA from stool for pathogen detection' and added 1 mL of Inhibit EX Buffer. A sterile 1 mL pipette tip was used to grind the sediment, and 0.5 mm sterile glass beads were added to help homogenize the sample. To reduce short DNA fragments, 0.4× AMPure XP beads were used. The quality and quantity of the obtained DNA were evaluated by running it in a 0.5% agarose gel and using the Qubit dsDNA assay kit (Thermo Fisher Scientific Inc.). Finally, DNA samples with a high molecular weight (modal size >2 kbp) and sufficient quantity (>10 $\mu$g) were used for sequencing.

**Library construction and sequencing.** For Illumina sequencing, a metagenomic shotgun sequencing library was constructed and sequenced at Shanghai Biozeron Biological Technology Co. Ltd. In brief, 1 $\mu$g of genomic DNA from each sample was sheared with a Covaris S220 Focused-ultrasonicator, and sequencing libraries with a fragment length of approximately 400 bp were prepared. The libraries were sequenced on a NovaSeq 6000 instrument (Illumina) in paired-end 150-bp mode. Finally, raw NGS reads were trimmed using the JAVA program Trimmomatic (version 0.33, www.usadellab.org/cms/?page=trimmomatic) to remove sequencing adapters and low-quality sequences (default parameters). The estimated coverage of the NGS and HiFi data was evaluated by using Nonpareil 3 with the 'kmer' option (63).

For PacBio sequencing, 5 $\mu$g of DNA was used to prepare a SMRTbell library with the PacBio SMRTbell prep kit 3.0 (Pacific Biosciences, Part Number: 102-182-700) according to the manufacturer's recommendations. Damaged double-stranded DNA in the initial DNA sample was repaired using the New England BioLabs PreCR Repair Mix Kit according to the manufacturer's instructions before library preparation. Then, the repaired DNA was size-selected by using the BluePippin system (Sage Science) to

obtain molecules larger than 3 kb. The SMRTbell library was sequenced with v3 chemistry on a PacBio Sequel IIe instrument (Pacific Biosciences) using SMRT 8M cells (Part Number: 101-389-001). A total of raw 359.80 Gb of long-read data were obtained, and HiFi reads were then generated with the 'ccs' module (parameters: –min-length 200 –min-passes 3 –min-rq 0.99) within the SMRT Link v10.0 package (Pacific Biosciences).

**HiFi, Hybrid, and NGS assembly processes.** In this study, we applied 3 strategies to assemble NGS and/ or HiFi reads. First, 'HiFi' mode involved 3 tools: HiCanu (https://github.com/marbl/canu, version 2.1, key parameters: 'genomeSize = 1000m maxMemory = 500 useGrid=false -pacbio-hifi') (64), HiFiasm-meta (https://github .com/lh3/hifiasm-meta, version 0.2-r053, default parameters) (33), and metaFyle (https://github.com/fenderglass/ Flye, version 2.8.1-b1676, key parameters: '–pacbio-hifi –meta -g 1g') (65). 'Hybrid' mode also involved 3 tools: MaSuRCA (https://github.com/alekseyzimin/masurca, version 4.0.3, corrected long-read mode with the default parameters) (66), hybridmetaSPAdes (https://github.com/ablab/spades, version 3.15.3, key parameters: '–meta – pacbio -m 500') (67), and Operams (https://github.com/CSB5/OPERA-MS, version: 2.11-r797, default parameters) (68). 'NGS' mode involved 2 tools, Megahit (https://github.com/voutcn/megahit, version: 1.1.1, parameters: '–min-contig-len 500 –k-min 21 –k-max 141') (69), and metaSPAdes (https://github.com/ablab/spades, version 3.15.3, key parameters: '–meta -m 500') (67). Contigs of less than 500 bp were discarded from all assembles.

**MAG construction and genome comparison.** The sequencing depth of each contig was calculated using the functional script 'jgi_summarize_bam_contig_depths', a tool of the MetaBAT2 (v.2.12.1) package (70), based on the sorted BAM files generated by using BWA-MEM (v.0.7.17, http://biobwa .sourceforge.net/) and SAMtools (v1.546, http://www.htslib.org/). Then, MetaBAT2 was applied to bin the assemblies with contig depth results under the default parameters (minimum contig length ≥1500 bp). CheckM (v.1.0.7) with the lineage_wf workflow was used to estimate the quality of MAGs (completeness and contamination), and retrieve the assembly information of MAGs (71). dRep was applied to cluster MAGs under an ANI > 95%, and the final representative MAGs with the highest quality score values (defined as completeness - 5X contamination) were selected (72). The GTDB Toolkit (GTDB-Tk, version r202) was introduced to obtain taxonomic information for each MAG (73). A Mummer plot comparing the MAGs and their closest genomes was drawn with the 'nucmer' and 'mummerplot' packages from MUMmer (https://github.com/mummer4/mummer, version 4.0), generally with the default options (74).

**Taxonomic classification and 16S/18S rRNA gene identification.** The taxonomy of NGS/HiFi reads was determined by Kraken2 (75) using the standard database (version: PlusPF-16, May 2021) with default parameters except for '–quick –report-zero-counts'. All reads were classified at 7 phylogenetic levels (domain, phylum, class, order, family, genus, species) or recorded as unclassified, and the abundances of different taxonomic groups were estimated with Bracken (https://ccb.jhu.edu/software/bracken/). *In silico*, 16S/18S rRNA gene sequences and completeness were extracted by using barrnap (https://github.com/ tseemann/barrnap, version 0.9). Different from the partial rRNA genes, the OTU clustering cutoff of the full-length rRNA genes should be higher than 97% and 98.65% was used as the clustering threshold in this study (76, 77). The 16S/18S-OTUs were clustered by using cd-hit-est with the following parameter option: -c 0.9865 -G 0 -M 0 -d 0 -aS 1 -r 1 (35, 78). Then, we annotated and aligned 16S/18S-OTUs against the Silva (Release 135, http://www.arb-silva.de) database by uclust algorithm within the usearch v11 software package (https://www.drive5.com/usearch/) with default parameters.

**BGC prediction.** The assemblies were used as the input for the BGC prediction tool antiSMASH (https://antismash.secondarymetabolites.org, version 5). BGCs were classified as 'Partial' (when they were found at a contig edge) or 'Complete' (when this was not the case) based on the annotated GenBank files. BGCs in which fewer than 50% of the genes showed hits with the best KnownClusterBlast hit was considered 'novel' BGCs, and others were defined as 'known' BGCs.

**Viral genome identification.** Contigs (≥10 kb) were used to predict viral genomes with VirSorter2 (https://github.com/jiarong/VirSorter2, version 2.2.3) (79). Hits with scores >0.9 and hallmarks >2 were considered positive viral genomes. Viral sequences sharing more than 95% nucleotide identity across more than 85% of the whole genome were dereplicated into groups, and the longest sequence within each group was chosen as the nonredundant viral operational taxonomic unit (vOTU) (80).

**Data availability.** The raw sequence read data analyzed in this study are available at China National GenBank (CNGB, https://db.cngb.org/) database under project number CNP0003352, while the accession numbers for the short-read and long-read data were CNX0487694 and CNX0487864, respectively.

## SUPPLEMENTAL MATERIAL

Supplemental material is available online only.
**SUPPLEMENTAL FILE 1**, PDF file, 0.6 MB.
**SUPPLEMENTAL FILE 2**, XLSX file, 0.2 MB.
**SUPPLEMENTAL FILE 3**, XLSX file, 0.1 MB.

## ACKNOWLEDGMENTS

We thank Peixin Gao for his help in the Tibet sampling champion.

This work was funded by the National Natural Science Foundation of China (31722008), the Second Tibetan Plateau Scientific Expedition and Research (STEP) program (Grant No. 2021QZKK0102 and 2019QZKK0503), Chinese Academy of Sciences (QYZDJ-SSW-DQC030), Science & Technology Basic Resources Investigation Program of China (2017FY100300), and the Youth Innovation Promotion Association of CAS (2014273).

P.X. and Q.L.W. designed the experiments. Y.T., F.X., C.Z., and B.L. performed the experiments. Y.T. analyzed the data and wrote the main manuscript. All authors read and approved the final manuscript.

We declare that there are no competing interests.

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
