## [Reviewer comments · Microbiology Spectrum]

Microbiology Spectrum

Improved assembly of MAGs and viruses in Tibetan saline lake sediment by HiFi metagenomic sequencing

Ye Tao, Fan Xun, Cheng Zhao, Zhendu Mao, Biao Li, Peng Xing, and Qinglong Wu

Corresponding Author(s): Peng Xing, Nanjing Institute of Geography and Limnology, Chinese Academy of Sciences

Review Timeline:

Submission Date:	August 26, 2022
Editorial Decision:	October 23, 2022
Revision Received:	November 17, 2022
Accepted:	November 22, 2022

Editor: Jinxin Liu

Reviewer(s): The reviewers have opted to remain anonymous.

Transaction Report:

DOI: <https://doi.org/10.1128/spectrum.03328-22>

October 23, 2022

Dr. Peng Xing
Nanjing Institute of Geography and Limnology, Chinese Academy of Sciences
Lake Ecology
73# East Beijing Road
Nanjing, Jiangsu 210008
China

Re: Spectrum03328-22 (Improved assembly of MAGs and viruses in Tibetan saline lake sediment by HiFi metagenomic sequencing)

Dear Dr. Peng Xing:

I apologize for the unnecessary delay in reaching a decision as I was having trouble collecting the review comments. some comments to consider from the editor to improve this work.

Do you think there is a sample preference for your assembling strategy? why or why not? Or why the authors chose Tibetan saline lake sediment for this analysis? This is important to generalize your approach in the microbiome community in general.

L123, "obtained 39.98 Gbp of paired-end short-read data and 9.17 Gbp of HiFi data", please explain what determined the sequencing data size. Is there any validation method? Like saturation, or some others.

Line 125-129: The authors' results show that the read assignment rate of HiFi is significantly higher than that of NGS. Is there a sequencing preference for high-abundance species in the samples, which increases the rate of HiFi? Is it reasonable to compare annotation rates between NGS and TGS given the large difference in overall data volume? In addition, the author should also explain the assignment rate at different taxonomic levels.

L156 and Table 1, Based on the same dataset, there are large differences in magnitude on multiple indicators of the assembly results. The authors should provide solid evidence or references to eliminate whether this is related to data or software parameters and algorithms.

Lines 508: Different parameter settings have a significant impact on the annotation results of kraken2, especially for NGS data. For example, --confidence, --quick and other parameters have a large impact on the results. The authors did not specify specific parameters in the method.

Line 137-139: Since a new sequence has been identified at the species level, how to judge whether the new sequence is complete?

Fig1: The image of NGS Novaseq 6000 display error.

Fig4: The number and completeness of contigs obtained using the 'Hybrid' assembly strategy to identify 16S/18S rRNA sequences also seem to be not as good as the 'HiFi' assembly strategy alone, can the reason for this be explained?

Fig4 and Fig7, the authors should provide the statistical method in detail.

Link Not Available

ASM policy requires that data be available to the public upon online posting of the article, so please verify all links to sequence records, if present, and make sure that each number retrieves the full record of the data. If a new accession number is not linked or a link is broken, provide production staff with the correct URL for the record. If the accession numbers for new data are not

publicly accessible before the expected online posting of the article, publication of your article may be delayed; please contact the ASM production staff immediately with the expected release date.

Sincerely,

Jinxin Liu

Journals Department
Reviewer comments:

Reviewer #2 (Comments for the Author):

The manuscript is interesting and will be of interest of the readers of the journal. The authors proposed a improved metagenome assembly strategy based on emerging high confident long-read sequencing technology compared with traditional short-read high-throughput sequencing mode. Using a Tibetan saline lake sediment as the demo sequencing sample, authors provided comprehensive HiFi metagenomic sequencing performances not only in MAG assembly, but also gave a picture for viral genomes and biosynthetic gene clusters identification with long read data. 'Hybrid (NGS + HiFi)' assembly strategy performed best by eight different popular meta-assemblers. This work is very interesting and has made a good attempt to combine short-read and long-read to improve the complex metagenome assembly quality in future. I would like to suggest to endorse this manuscript for a publication in microbiology spectrum. However there is few comments that need to be addressed.

L127-128 The taxonomic assignment rate of HiFi data was 90.8% which was much higher than short read data. Were most of long reads classified at specie level, or not? Please clarify and provide evidence to support this result.

L138 Authors only focus on the complete 16S/18S rRNA genes. How to determine the intact of 16S/18S genes? Were there partial rRNA genes from novel species? Please provide more details in this part because this was one of the greatest advantages using HiFi long reads.

L140 For OTU identification 97% identity was always the default parameters, why authors used 98.65% as the cutoff? Please provide more citations.

L146-148 Two abundant 18S-OTUs were unknown which was very strange here (Figure 2B). Authors did not mention the tool and parameters to annotate the OTUs (Line 516).

L160-166 (Table 1) There were huge differences for the total contig sizes of 'HiFi' assemblers, especially for HiCanu, although authors found that high abundance species were successfully retrieved. Please explain this interesting phenomenon.

L232 Figure 5B, 'Hybrid' strategy generated the most number of representative MAGs. I was very curious about the reason why 'Hybrid'-specific MAGs could not be retrieved from another two strategies. Please give more details about these kinds of MAGs.

L241-243 Please clarify the principle of 'HiFi' assemblers to judge the topology of contigs. Is it necessary to combine other data to support? The same question about the circular viral genome (Line 295).

L240 Authors provided two cMAG and compared them with published known genomes in this study. But I can not figure out the advantages of 'HiFi' data and here lack of the comparisons with the corresponding incomplete MAGs assembled by other methods.

Staff Comments:

Preparing Revision Guidelines

Please return the manuscript within 60 days; if you cannot complete the modification within this time period, please contact me. If you do not wish to modify the manuscript and prefer to submit it to another journal, please notify me of your decision immediately so that the manuscript may be formally withdrawn from consideration by Microbiology Spectrum.

Comments to the Author

The manuscript is interesting and will be of interest of the readers of the journal. The authors proposed a improved metagenome assembly strategy based on emerging high confident long-read sequencing technology compared with traditional short-read high-throughput sequencing mode. Using a Tibetan saline lake sediment as the demo sequencing sample, authors provided comprehensive HiFi metagenomic sequencing performances not only in MAG assembly, but also gave a picture for viral genomes and biosynthetic gene clusters identification with long read data. 'Hybrid (NGS + HiFi)' assembly strategy performed best by eight different popular meta-assemblers. This work is very interesting and has made a good attempt to combine short-read and long-read to improve the complex metagenome assembly quality in future. I would like to suggest to endorse this manuscript for a publication in microbiology spectrum. However there is few comments that need to be addressed.

L127-128 The taxonomic assignment rate of HiFi data was 90.8% which was much higher than short read data. Were most of long

reads classified at specie level, or not? Please clarify and provide evidence to support this result.

L138 Authors only focus on the complete 16S/18S rRNA genes. How to determine the intact of 16S/18S genes? Were there partial rRNA genes from novel species? Please provide more details in this part because this was one of the greatest advantages using HiFi long reads.

L140 For OTU identification 97% identity was always the default parameters, why authors used 98.65% as the cutoff? Please provide more citations.

L146-148 Two abundant 18S-OTUs were unknown which was very strange here (Figure 2B). Authors did not mention the tool and parameters to annotate the OTUs (Line 516).

L160-166 (Table 1) There were huge differences for the total contig sizes of 'HiFi' assemblers, especially for HiCanu, although authors found that high abundance species were successfully retrieved. Please explain this interesting phenomenon.

L232 Figure 5B, 'Hybrid' strategy generated the most number of representative MAGs. I was very curious about the reason why 'Hybrid'-specific MAGs could not be retrieved from another two strategies. Please give more details about these kinds of MAGs.

L241-243 Please clarify the principle of 'HiFi' assemblers to judge the topology of contigs. Is it necessary to combine other data to support? The same question about the circular viral genome (Line 295).

L240 Authors provided two cMAG and compared them with published known genomes in this study. But I can not figure out the advantages of 'HiFi' data and here lack of the comparisons with the corresponding incomplete MAGs assembled by other methods.

Responses to editor's and reviewers' comments:

To editor,

We are very grateful to the two anonymous reviewers for their contributions to the manuscript. We have considered the comments and suggestions and made some revisions as follows.

Reviewer #1 (Comments for the Author):

Comments to the Author

Do you think there is a sample preference for your assembling strategy? why or why not? Or why the authors chose Tibetan saline lake sediment for this analysis? This is important to generalize your approach in the microbiome community in general.

Reply: Thank you very much for your comments and suggestions.

Up to recently, PacBio high-fidelity (HiFi) sequencing technology has been used in the metagenomic research area mainly focusing on the gut microbiome such as human (Kim *et al.* 2022) and sheep fecal (Bickhart *et al.* 2022) samples. Taking into account that the HiFi sequencing method is an emerging technology, there is no widely accepted assembling strategy. For example, Kim *et al.* tried three different tools to assemble human fecal samples, and the number of contigs varied from 481 to 2,283. In our manuscript, we also found the same phenomenon that there were huge differences among assemble results from different tools. Therefore, there is no sample preference for our assembling strategy.

The reasons why we chose a Tibet saline lake in this study, first were that no studies have yet used HiFi for metagenome sequencing of non-intestinal microorganisms. We can make a meaningful attempt for environmental samples. The second was that our lab had cultivated some isolates from saline lake sediments which could be used to do genome comparison with the MAGs. Obviously, due to the exceeding complexity of environmental samples, our results could only provide a feasible attempt for HiFi technology in environmental metagenomic study.

Reference

- Bickhart DM, Kolmogorov M, Tseng E, Portik DM, Korobeynikov A, Tolstoganov I, Uritskiy G, Liachko I, Sullivan ST, Shin SB, Zorea A, Andreu VP, Panke-Buisse K, Medema MH, Mizrahi I, Pevzner PA, Smith TPL. 2022. Generating lineage-resolved, complete metagenome-assembled genomes from complex microbial communities. *Nat Biotechnol* 40(5):711-719. <https://doi.org/10.1038/s41587-021-01130-z>.
- Kim CY, Ma J, Lee I. 2022. HiFi metagenomic sequencing enables the assembly of accurate and complete genomes from human gut microbiota. *Nat Commun* 13(1):6367. <https://doi.org/10.1038/s41467-022-34149-0>.

L123, "obtained 39.98 Gbps of paired-end short-read data and 9.17 Gbps of HiFi data", please explain what determined the sequencing data size. Is there any validation method? Like saturation, or some others.

Reply: Thank you very much for your comments and suggestions.

For short-read data, we got 133,280,744 raw read pairs (CNGB accession number: CNX0487694) with 150 bp length which meant about 39.98 Gbps ($133,280,744 * 2 * 150 \text{ bp} = 39.98 \text{ Gbps}$). A nonpareil pipeline was introduced to estimate the data saturation (Figure S9A).

For long-read data, we got 2,033,496 HiFi reads (CNGB accession number: CNX0487864, 9.17Gbp) from the Sequel II instrument by SMRT Link v10.0 package. We added 16S/18S-OTU saturation curves here to demonstrate the HiFi data advantage in full-length complete rRNA genes identification.

Line 125-129: The authors' results show that the read assignment rate of HiFi is significantly higher than that of NGS. Is there a sequencing preference for high-abundance species in the samples, which increases the rate of HiFi? Is it reasonable to compare annotation rates between NGS and TGS given the large difference in overall data volume? In addition, the author should also explain the assignment rate at different taxonomic levels.

Reply: Thank you very much for your comments and suggestions.

The taxonomy of NGS/HiFi reads was determined by Kraken2 whose classification algorithm was based on the *k*-mer of reads matching known genomes with the lowest common ancestor (LCA) methods (Wood *et al.* 2014). The known genomes were from the standard database (version: PlusPF-16, May 2021) of Kraken2 and all genomes were de-duplicated according to the taxonomic species level. Thus, there is no preference for high-abundance species in the samples. Because of more *k*-mer information exists in HiFi long-read data than short-read data within a single read, it is obvious that HiFi data have a significantly higher assignment rate than NGS data. The same phenomenon was also reported by Portik *et al.* 2022.

Reference

Wood DE, Salzberg SL. 2014. Kraken: ultrafast metagenomic sequence classification using exact alignments. *Genome Biol* 15(3):R46. <https://doi.org/10.1186/gb-2014-15-3-r46>.

Portik DM, Brown CT, Pierce-Ward NT. 2022. Evaluation of taxonomic profiling methods for long-read shotgun metagenomic sequencing datasets. bioRxiv. <https://doi.org/10.1101/2022.01.31.478527>

L156 and Table 1, Based on the same dataset, there are large differences in magnitude on multiple indicators of the assembly results. The authors should provide solid evidence or references to eliminate whether this is related to data, software parameters, and algorithms.

Reply: Thank you very much for your comments and suggestions.

Yes, there are large differences in assembly results generated by eight different tools. Taking into account that the HiFi sequencing method is an emerging technology, there is no widely accepted software to assemble HiFi reads, let alone combine short- and long-read. In this study, we tried our best to use eight different tools to assemble the corresponding data. Kim *et al* tried three different tools to assemble human fecal samples, and the number of contigs varied from 481 to 2,283. Up to now, it is lack of papers to evaluate the metagenomic assemblers supporting long-read data systematically. We had added more results in the revised manuscript.

Lines 508: Different parameter settings significantly impact the annotation results of kraken2, especially for NGS data. For example, --confidence, --quick and other parameters have a large impact on the results. The authors did not specify specific parameters in the method.

Reply: Thank you very much for your comments and suggestions.

In the revised manuscript, we added more details about software parameters in the methods section.

Line 137-139: Since a new sequence has been identified at the species level, how to judge whether the new sequence is complete?

Reply: Thank you very much for your comments and suggestions.

For the completeness of 16S/18S rRNA genes, we got these results from *gff* files generated by barrnap (<https://github.com/tseemann/barrnap>, version 0.9). rRNA prediction results with a 'partial' tag in *gff* files were discarded in this study.

Fig1: The image of NGS Novaseq 6000 display error.

Reply: Thank you very much for your comments and suggestions.

We added the right image of Novaseq 6000 in the revised manuscript.

Fig4: The number and completeness of contigs obtained using the 'Hybrid' assembly strategy to identify 16S/18S rRNA sequences also seem to be not as good as the 'HiFi' assembly strategy alone, can the reason for this be explained?

Reply: Thank you very much for your comments and suggestions.

Three assemblers were used to do 'Hybrid' assembly including Operams, HybrdimetaSPAdes, and MaSuRCA, but the corresponding algorithms were different. Operams and HybrdimetaSPAdes assembled short-read data by Megahit and metaSPAdes respectively and then ordered and expanded contigs using long-read data. The second step could not significantly improve the completeness of genes. Therefore, the 16S/18S rRNA genes seem

not as good as the 'HiFi' assembly results. Moreover, MaSuRCA was first designed to assemble '454' data whose read length was around 500 bp and was based on the overlap-based assembly algorithm. This strategy is similar to the 'HiFi' overlap-based assemblers. The completeness of 16S/18S rRNAs from MaSuRCA is closer to that generated from the 'HiFi' assembly results.

Fig4 and Fig7, the authors should provide the statistical method in detail.

Reply: Thank you very much for your comments and suggestions.

We added the statistical method in the figure legend in the revised manuscript.

Reviewer #2 (Comments for the Author):

The manuscript is interesting and will be of interest to the readers of the journal. The authors proposed a improved metagenome assembly strategy based on emerging high confident long-read sequencing technology compared with traditional short-read high-throughput sequencing mode. Using a Tibetan saline lake sediment as the demo sequencing sample, authors provided comprehensive HiFi metagenomic sequencing performances not only in MAG assembly, but also gave a picture for viral genomes and biosynthetic gene clusters identification with long read data. 'Hybrid (NGS + HiFi)' assembly strategy performed best by eight different popular meta-assemblers. This work is very interesting and has made a good attempt to combine short-read and long-read to improve the complex metagenome assembly quality in future. I would like to suggest to endorse this manuscript for a publication in microbiology spectrum. However there is few comments that need to be addressed.

L127-128 The taxonomic assignment rate of HiFi data was 90.8% which was much higher than short read data. Were most of long reads classified at specie level, or not? Please clarify and provide evidence to support this result.

Reply: Thank you very much for your comments and suggestions.

67.19% of long reads and 5.59% of short reads could be classified at the species level, respectively. We added more details about this part in the revised manuscript.

L138 Authors only focus on the complete 16S/18S rRNA genes. How to determine the intact of 16S/18S genes? Were there partial rRNA genes from novel species? Please provide more details in this part because this was one of the greatest advantages using HiFi long reads.

Reply: Thank you very much for your comments and suggestions.

We got the completeness extent of the 16S/18S rRNA genes from *gff* files generated by barnap (<https://github.com/tseemann/barnap>, version 0.9). The rRNA prediction results with a 'partial' tag in *gff* files were discarded in this study. One of the greatest advantages of HiFi reads was to extract complete rRNA efficiently. And we did 16S/18S-OTU saturation curves here to validate the HiFi data in full-length complete rRNA genes identification. Therefore, it is not necessary to analyze partial rRNA genes.

L140 For OTU identification 97% identity was always the default parameters, why authors used 98.65% as the cutoff? Please provide more citations.

Reply: Thank you very much for your comments and suggestions.

97% identity threshold was always used in short-read-based 16S rRNA OTUs clustering. All of these kinds of data were incomplete 16S rRNA genes. For the complete 16S rRNA gene, a 98.65% cutoff was always applied (Edgar *et al.* 2018; Johnson *et al.* 2019; Gao *et al.* 2022). More citations were added in the revised manuscript.

Reference

- Edgar RC. 2018. Updating the 97% identity threshold for 16S ribosomal RNA OTUs. *Bioinformatics* 34(14):2371-2375. <https://doi.org/10.1093/bioinformatics/bty113>.
- Johnson JS, Spakowicz DJ, Hong BY, Petersen LM, Demkowicz P, Chen L, Leopold SR, Hanson BM, Agresta HO, Gerstein M, Sodergren E, Weinstock GM. 2019. Evaluation of 16S rRNA gene sequencing for species and strain-level microbiome analysis. *Nat Commun* 10(1):5029. <https://doi.org/10.1038/s41467-019-13036-1>.
- Gao F, Guo R, Ma Q, Li Y, Wang W, Fan Y, Ju Y, Zhao B, Gao Y, Qian L, Yang Z, He X, Jin X, Liu Y, Peng Y, Chen C, Chen Y, Gao C, Zhu F, Ma X. 2022. Stressful events induce long-term gut microbiota dysbiosis and associated post-traumatic stress symptoms in healthcare workers fighting against COVID-19. *J Affect Disord* 303:187-195. <https://doi.org/10.1016/j.jad.2022.02.024>.

L146-148 Two abundant 18S-OTUs were unknown which was very strange here (Figure 2B). The authors did not mention the tool and parameters to annotate the OTUs (Line 516).

Reply: Thank you very much for your comments and suggestions.

Usearch 'uclust' algorithm was applied to annotate the taxonomy of 18S-OTUs. A detailed annotation description was added in the methods section of the revised manuscript.

L160-166 (Table 1) There were huge differences for the total contig sizes of 'HiFi' assemblers, especially for HiCanu, although authors found that high abundance species were successfully retrieved. Please explain this interesting phenomenon.

Reply: Thank you very much for your comments and suggestions.

The reasons why the sizes of assembles varied dramatically were: (1) HiFi sequencing method was an emerging technology and there is no widely accepted software to assemble HiFi reads, let alone combine short- and long-read. In this study, we tried our best to use eight different tools to assemble the corresponding data. (2) Up to now it is a lack of papers to evaluate the metagenomic assemblers supporting long-read data systematically. Kim *et al.* tried three different tools to assemble human fecal samples, and the number of contigs varied from 481 to 2,283 coinciding with our results. HiCanu again generated the smallest assembly indicating that HiCanu might need more HiFi data. (3) HiFi data currently was much more expensive than short-read data, and we only generated 1 cell for the CCL sediment sample. We did saturation curves to validate the HiFi data in full-length complete rRNA genes, but to a certain degree, it was obvious that 9.17 Gbps HiFi data was not sufficient for all three 'HiFi' assemblers.

L232 Figure 5B, 'Hybrid' strategy generated the most number of representative MAGs. I was very curious about the reason why 'Hybrid'-specific MAGs could not be retrieved from another two strategies. Please give more details about these kinds of MAGs.

Reply: Thank you very much for your comments and suggestions.

We checked the status of 'Hybrid'-specific MAGs in 'HiFi' and 'NGS' MAG results. A total of 13 'Hybrid'-specific MAGs were not binned successfully within 'HiFi' contigs indicating these MAGs could not be retrieved under low-depth HiFi data. On the other hand, these 13 'Hybrid'-specific MAGs were initially binned across two 'NGS' results, however, were filtered because of the low genome completeness. These results demonstrated the great advantages of MAG construction as a 'Hybrid' strategy.

L241-243 Please clarify the principle of 'HiFi' assemblers to judge the topology of contigs. Is it necessary to combine other data to support? The same question about the circular viral genome (Line 295).

Reply: Thank you very much for your comments and suggestions.

HiCanu, HiFiasm-meta, and metaFyle are the typical long-read assemblers nowadays. All of them could provide the genome topology of contigs, but it is a pity that the judge cutoffs for circular genomes could not be found in the corresponding software papers.

L240 Authors provided two cMAG and compared them with published known genomes in this study. But I can not figure out the advantages of 'HiFi' data and here lack of comparisons with the corresponding incomplete MAGs assembled by other methods.

Reply: Thank you very much for your comments and suggestions.

In the 'Complete MAG (cMAG) assembly ability' section, we compared two cMAGs with published known genomes. Total medium/high-quality MAGs generated by different strategies were compared together in Figure 4. We found two important advantages of HiFi data in metagenome studies under the near amount of long-read data. The first one was to identify the full-length 16S/18S rRNA genes that would not be efficiently retrieved by other sequencing data. The second one was that combining short-read data, the 'Hybrid' assembly strategy could increase the quantity and quality of MAGs. Of course, without any doubt, long-read sequencing technology could provide more complete and accurate MAGs from environmental samples as the HiFi costs continue to fall in the future.

November 22, 2022

Dr. Peng Xing
Nanjing Institute of Geography and Limnology, Chinese Academy of Sciences
73# East Beijing Road
Nanjing, Jiangsu 210008
China

Re: Spectrum03328-22R1 (Improved assembly of MAGs and viruses in Tibetan saline lake sediment by HiFi metagenomic sequencing)

Dear Dr. Peng Xing:

Your manuscript has been accepted, and I am forwarding it to the ASM Journals Department for publication. You will be notified when your proofs are ready to be viewed.

Sincerely,

Jinxin Liu
Editor, Microbiology Spectrum

Journals Department
Supplemental Material: Accept
Supplemental Material: Accept
Supplemental Material: Accept